# Geopolitics of Catholic Pilgrimage: On the Double Materiality of (Religious) Politics in the Virtual Age

**Petr Kratochvíl**

Institute of International Relations, 118 00 Prague, Czech Republic; kratochvil@iir.cz

**Abstract:** This article explores geopolitical aspects of Catholic pilgrimage in Europe. By exploring the representations of pilgrimage on Catholic social media, it shows that the increasing influence of the virtual is accompanied by a particular reassertion of the material aspects of pilgrimage. Two types of Catholic pilgrimage emerge, each with a particular spatial and political orientation. The first type of pilgrimage is predominantly politically conservative, but also spatially static, focusing on objects, be they human bodies or sacred sites. The second type is politically progressive, but also spatially dynamic, stressing pilgrimage as movement or a journey. The classic Turnerian conceptualization of a pilgrimage as a three-phase kinetic ritual thus falls apart, with liminality appropriated by the progressive type and aggregation almost entirely taken over by the conservative, apparitional pilgrimage. As a result, pilgrimage has once again become a geopolitical reflection of the broader ideological contestation both within Christianity and beyond.

**Keywords:** geopolitics; Catholic Church; Europe; pilgrimage; materiality; politics; ideology; Virgin Mary

## 1. Introduction

"You cannot do a pilgrimage on your couch" was the message of the rector of the Lourdes sanctuary, which he recently posted on YouTube (KTOTV 2020a). The paradox of the insistence on the material aspects of pilgrimage and the simultaneous ever-growing virtualization of religious pilgrimage is difficult to ignore. In the context of the Catholic Church alone, hundreds of virtual pilgrimage tours are offered on the Internet, and pilgrimage videos and documentaries belong among the most frequently posted videos on religious websites. The pilgrimage centers also follow the trend, encouraging e-pilgrimages and offering hybrid forms of vicarious pilgrimage, which combine physical and virtual elements. The coronavirus pandemic further accelerated the trend towards virtualization. The online attendance of pilgrimages skyrocketed, with a 5-fold increase in virtual attendance of Lourdes (Virtual Pilgrimage 2020). Moreover, online transmissions of the rituals at pilgrimage centers have become a staple of religiously oriented YouTube channels and Facebook sites.

However, the more virtual pilgrimages flourish and the more social media are employed, the more the physical elements seem to be stressed. No matter how popular virtual pilgrimage tours have become, the importance of a "real" or "proper" pilgrimage is repeatedly underlined. The heightened attention to the material, however, takes different forms. Some pilgrimages focus on the human body, on its physical healing, and on corporeality in general; others tend to see pilgrimage as a walk through a landscape, and yet others look for parallels between movement through space and the temporal movement of human life. All of these share one thing in common, though. The spatiality of pilgrimage, no matter how diverse it is, is always also political. The diversification of pilgrimage types means also a diversification of political claims about what the church is and how it should relate to the late modern world of today. In short, pilgrimage has become a geopolitical reflection of the ideological contestation both within Christianity and beyond.

The transformation and diversification of pilgrimage have serious consequences for the way we understand the phenomenon of pilgrimage as such. In their classic

account of pilgrimage, Turner and Turner (Turner and Turner 2011) describe pilgrimage as a ritual consisting of three phases—*separation*, *limen*, and *aggregation*. Pilgrims remove themselves from their ordinary, secular social order (separation); experience the limits of the normal order and the possibilities for its transcendence (liminality); and are, in the final phase, reintegrated in the community (aggregation). While the three phases were always represented in any pilgrimage in varying ratios, all three were nonetheless always present. However, as this article will try to show, the recent evolution of pilgrimage in Europe challenges this received wisdom as different types of pilgrimage prioritize one dimension, marginalizing or entirely suppressing the others. The impact of the fragmentation is substantial as different types of pilgrimages now not only serve different purposes for the pilgrims, but their theopolitical orientations also start to radically differ.

## 2. The Late Modern Revival of Catholic Pilgrimage

The revival of pilgrimage in Europe is, in a sense, both a product of Europe's secularization and a reaction to it. There is no doubt that Europe remains the most secularized part of the world (Martin 2017; but see the discussion in Berger 2002 and in Voas and Chaves 2016). However, while secularization in terms of organized religious practice continues to decline in all parts of Europe (Molteni and Biolcati 2018), pilgrimage enjoys an unprecedented level of growth, both in terms of intra-European pilgrimage and in terms of travel to sacred places elsewhere (Reader 2007; Albera and Eade 2016). Pilgrimage is thus an area that shows that the secularization process in Europe is uneven and that particular religious/spiritual activities are becoming more popular, the general trend notwithstanding.[1]

The reasons for the growing popularity of pilgrimage are manifold, and only some of them are related to traditional Catholic religiosity. While Catholic media typically refer to the pilgrimages to the Marian sites, that is, to the "apparitional" type of pilgrimage (e.g., pilgrimages to Fátima, Lourdes, or Međugorje), the spectrum of pilgrimages is substantially wider. Importantly, pilgrimages are part of the burgeoning religious tourism (Nolan and Nolan 1992), and many pilgrims exhibit attitudes closer to secular tourists than to devout Christians (cf. Collins-Kreiner and Kliot 2000). Some traditional Christian pilgrimages such as the Santiago camino have become popular with yet another segment of the European population which is skeptical both of organized religious traditions and of pilgrimage tourism (Reader 2007).

The reappropriation of the old, medieval pilgrimage (such as the above-mentioned Way of St. James) by people of various non-Catholic spiritualities was made possible by the new focus of the Catholic pilgrimage itself. The rise of the mass "apparitional" pilgrimage in Europe is a relatively novel phenomenon, as it was only in the 19th and 20th centuries when the Marian devotion started to play a primary role in Catholic European piety. "The Age of Mary"[2] brought attention to the Virgin Mary not only as the Theotokos of old, but as a much more dynamic figure actively intervening in human affairs (Turner and Turner 1982). The reorientation to apparitional pilgrimage and its mass character resulted in a dramatic redefinition of what a Catholic pilgrimage is about and what its destinations are. As a consequence, while new Catholic pilgrimage centers continued to appear in the last two centuries, the vast majority of those were related to Marian devotion. Medieval pilgrimage centers then either lost their once immense popularity (Cologne, Canterbury) or redefined themselves in novel ways (Rome, Santiago de Compostela).

## 3. Two Variants of Pilgrimage, Two Variants of Catholicism

The rise of the pilgrimage centered upon the Virgin Mary is connected to the specific political and economic context of 19th century Europe.[3] Urbanization, industrialization, and the growth of socioeconomic disparities are among the key background factors against which the Marian pilgrimage gained its modern contours. The 19th century Marian devotion and pilgrimage movement were thus characterized not only by "fervent personal piety" (Turner and Turner 1982, p. 18), but also by their apologetical attitude vis-à-vis the emerging liberal secularism. This latter feature became one of the key distinguishing aspects

of the Marian pilgrimage, and, in the end, this pilgrimage type became the ultimate popular embodiment of the 19th century ecclesia militans, a response of the embattled church to the changing world, and a bastion against the errors of modernism and liberalism (Margry 2009a; Sorrel 2016). The struggle of the church against its earthly and supernatural enemies often gained an explicit political dimension, such as in one of Fátima's hidden secrets which focused on the then atheist Russia (cf. the excellent analysis in Warner 2013, 320ff).

Correspondingly, the theological underpinnings of the Marian pilgrimage were also closely related to the popular Catholic theology of the 19th and the first half of the 20th century. The visions of Catherine Labouré,[4] the miraculous medal, the famous Lourdes song "Ô Vierge Marie", and the veneration of the immaculate heart (made famous in Fátima) are not only central for Marian pilgrimages, but also form the popular underpinning of Catholic Mariology and the dogma pronounced in 1854. To sum up, the "apparitional" Marian pilgrimage has not only become the most popular kind of Catholic pilgrimage worldwide, but in Europe, it also turned into the most visible popular reaction against modern secularization. Interestingly, the Marian pilgrimage retained many of these political features for the entire 20th century and beyond. The binary opposition between faith and the idolatrous world all but further sharpened in many pilgrimage centers in the course of the 20th century. A fervent anticommunism then became one of the hallmarks of Marian devotion and further stressed the duality of Catholic piety vs. the depravity of the godless (see, for instance, Donnelly 2005).

Interestingly, the Catholic Church has often been, on the official level, rather skeptical of some aspects of the Marian pilgrimage. The Second Vatican Council, while integrating popular religiosity in its liturgical reforms, tried to de-emphasize Mariology, which resulted in partially curbing the zeal of the popular Marian devotion (Preston 2017). The Christological focus of the Council and its rejection of "externalism" were also interpreted as blows targeting Marian devotion (Donnelly 2005). The convenor of the Council, Pope John XXIII, was openly skeptical towards some elements of the conservatively oriented pilgrimage movement, and he fought actively against the popular pilgrimage to San Giovanni Rotondo (Di Giovine 2011, p. 252).[5] Some tension between the spirit of the Council and popular Marian pilgrimage persists even today. The publications (including those by Catholic authors) that are critical of some Marian pilgrimage sites and question their authenticity often argue that their appeal is due to the confusion of many Catholics following the Second Vatican Council (Medjugorje's Mystery 2006; see also Foley 2002).

However, a second, less conservative type of pilgrimage developed in parallel with the rise of the Marian pilgrimage. This second type emerged as a result of a coalescence of several alternative spiritual movements. One of the sources was the secularized version of pilgrimage in the form of trips to the ancient civilizational centers of Europe, Rome in particular. The British *Grand Tour* of the 17th and 18th centuries is a typical example thereof. While outwardly, this kind of travel was decidedly nonreligious, it still bore a close relationship with pilgrimage, while also serving spiritual purposes (Colleta 2015, p. 74). In the second half of the 20th century, a second element of this less church-centered pilgrimage movement gained importance: the spirituality of ecumenical cooperation among Christian churches and of the universal brotherhood and sisterhood of all people. Ecumenical communities such as that of Taizé have contributed to the reinvention of this less institutionalized and less hierarchy-dependent form of Christian pilgrimage (Vilaça 2010). This form of pilgrimage became more popular not only in the Christian ecumenical milieu, but also beyond it. Those projects which transcend the boundary of Christianity and focus on interreligious dialogue, such as Hans Küng's global ethics, have had a very similar effect on the intellectual perception of pilgrimage as a general expression of every human's spiritual journey (Küng 2006). Finally, the above-mentioned suspicious attitude of the Catholic hierarchy towards some forms of traditional Marian pilgrimage and the general reorientation of the Catholic Church towards a more accepting attitude toward the modern world (*aggiornamento*) have had a decisive impact on both of

the types of pilgrimage discussed here, deflecting the attention from the first and indirectly encouraging the second.

Having said this, it is important to note that the two types of pilgrimage described here are ideal types, and many mixed forms exist in the grey zone between the two positions. In particular, pilgrimages to Rome and Jerusalem often contain elements of both, and unlike in the cases of Fátima and Međugorje, the likely ideological orientation of a particular group of pilgrims cannot be derived from the destination alone. The pilgrims on the *camino* to Santiago also show elements of both types of spiritualities (Vilaça 2010). The same caveat applies to the Marian devotees themselves. While the conservative features of the Marian *cultus* remain dominant, Marian devotion is expressed in increasingly diverse forms. Some of these are more sympathetic to the modern world, focusing on a reconciliation with it. Others even include a stress of gender emancipation with the Virgin serving as a role model (see, for instance, Margry 2009b).

## 4. Pilgrimage and Materiality

These two types of Catholic pilgrimage do not, however, serve only as ideal types. Specific pilgrimage sites and routes are associated with one, and other pilgrimage destinations with the other. Fátima, Lourdes, and Međugorje, to name some important European pilgrimage destinations, are clearly connected with the conservative, apparitional type of pilgrimage. Others, most notably, Taizé, but also increasingly Santiago de Compostela, are connected with the progressive type. Interestingly, a number of studies have shown that this difference does not only pertain to the type of theology propounded at the site. The pilgrimage sites are visited by different segments of the Catholic Church, with different expectations and with different aims (Vilaça 2010; Pack 2010).

Pilgrimage is not only spatially differentiated, though. It is spatial in a deeper sense as well. Pilgrimage is essentially "a kinetic ritual" (Turner and Turner 2011, p. xiii), and the transformative element of pilgrimage is not only related to individual conversion, but it is also fundamentally a "spatiotemporal social change" (Turner and Turner 2011, p. 2). Spatiality and more broadly materiality are a key element of pilgrimage, as a movement of spirit and *body* through space. The question is how far recent societal changes pose a serious challenge to this essential aspect of pilgrimage. One such challenge is the virtualization of pilgrimage as a consequence of the growing popularity of the virtual space in general and social media in particular. The recent coronavirus pandemic is another, more recent, but equally profound challenge.

The key questions this article wants to explore are thus related to these developments: Is the materiality of pilgrimage still relevant in spite of the virtualization of the social (as well as spiritual) life of today's European societies? If so, do the two different types of pilgrimage approach materiality differently? Finally, what is the geopolitics of pilgrimage, including both the political contestation within the church and the broader positioning of the church in Europe's late modern politics?

The article offers answers to these questions based on a detailed analysis of some of the most popular Catholic social media outlets and their depiction of pilgrimage. Four popular Catholic Facebook groups were analyzed: Catholic and Proud (6.0 million followers), Catholic Online (3.1 million followers), I Love My Catholic Faith (2.6 million followers), and Catholic Church (2.6 million followers). Additionally, three Catholic YouTube channels with a broad audience were explored: the Eternal Word Television Network (EWTN, 515,000 subscribers), KTO télévision catholique (434,000 subscribers), and Rome Reports en Español (346,000 subscribers). While the Facebook groups were predominantly oriented towards English speakers, the YouTube channels were intentionally chosen to represent a more diverse audience, as they cover not only English language news, but also news in Spanish (Rome Reports) and French (KTO). In each case, the posts/statuses/videos that contained the key words related to pilgrimage were collected ("pilgrim", "pilgrims", and "pilgrimage" in English; "peregrino", "peregrinos", and "peregrinaje" in Spanish; and "pèlerin", "pèlerins", and "pèlerinage" in French), and the sample covered the period

of January–December 2020. Altogether, 232 Facebook posts (including videos or texts from other websites which were hyperlinked in these posts) and 146 YouTube videos were collected and analyzed.[6] The analysis was conducted in two rounds. First, the two types were identified and analyzed in a small sample. Second, this analysis was extended to the entire corpus of texts and videos.

A final, terminological caveat is due here. I position my research within the broader field of the geopolitics of religion, which is part of the broader tradition of critical geopolitics. While geopolitics deals with the relationship between space and politics, its classical version explores (a) the impact of space on politics (and not vice versa) and (b) the relations among the traditional actors of global politics, i.e., sovereign states. My critical understanding of geopolitics is different in both these dimensions. Critical geopolitics of religion deals (a) with the impact of space on politics, but also of politics on spatial representations and practices and (b) with the activities of nonstate entities such as churches, religious communities, or movements.

## 5. The Double Geopolitics of Pilgrimage

One might expect that the growing popularity of social media and the onset of the coronavirus pandemic would modify the overall narrative, including the very definition of what a pilgrimage is and to what extent a pilgrimage has to contain material elements. Indeed, virtual pilgrimages have become a popular genre on the Catholic YouTube channels, and so have regular transmissions of masses and prayers via social media. Paradoxically, this virtualization has had the opposite effect of reducing the importance of the materiality of a pilgrimage. While previously, materiality was taken for granted, in 2020, the insistence on the physical aspects of pilgrimage became one of the hallmarks of pilgrimage-related texts and videos.

The virtual pilgrimages, while sometimes seen as useful props, started to be frequently compared to "proper" pilgrimages. Strongly worded statements such as that taking part in a pilgrimage online "will never replace proper physical experience" (EWTN 2020e) have become commonplace. A video from September 2020 where the Pope himself speaks about the joyful experience of a face-to-face meeting as opposed to a "screen-to-screen" meeting was widely shared and watched (ROME REPORTS en Español 2020). The online participation in a pilgrimage continued to be encouraged, but the asymmetry was firmly established: meeting "in spirit" serves only as a consolation for those who cannot be "really" present (EWTN 2020a). The expression "vicarious pilgrimage" started to be used frequently. It not only denoted the substitutionary dimension of the pilgrimage of somebody who cannot physically take part, but also reaffirmed the hierarchy between the proper pilgrimage and the vicarious type (EWTN 2020d). However, what the "proper" pilgrimage was and what constituted its materiality were expressed substantially differently in each type of pilgrimage. These different materialities brought with themselves also different political orientations, giving rise to two markedly different geopolitics of pilgrimage.

## 6. Corporeality and Its Absence in Conservative/Progressive Pilgrimage

The first distinguishing feature is the focus on corporeality in the conservative, Marian pilgrimage and the lack thereof in the progressive type. The reports on the conservative type insisted on the importance of a physical pilgrimage, on "honoring Mary in a physical manner" (Benedict 2020a). However, the stress on corporeality distinguishes the apparitional pilgrimage from all else. In particular, the physical healing of the bodies of the pilgrims was always a central element of the Marian pilgrimage narrative, in particular in Lourdes. This element gained further traction during the coronavirus pandemic as the healing waters of Lourdes were seen as an effective remedy against the virus—a universal healing source against a universal virus. "Let every believer touch the healing water please. So that through their belief, they will be cured" was a comment below a report on the closure of Lourdes (Catholic and Proud 2020b). Statuses and articles connecting pilgrimage with bodily health and the effects of the faith on physical renewal were very common.

Often, the corporeality was stressed to the minute detail. A pilgrimage-related article, which was shared on a conservative website, insisted that the Lent period is a pilgrimage that also cleanses our body: "Floss your teeth . . . your gums need a little extra and this will make your mouth feel NEW again" (Weiss 2018). Several other corporeal topics were frequently discussed in connection with the conservative type of pilgrimage: one such is the miraculous medal and the importance of wearing the medal on the believer's body so that its healing power can take effect (EWTN 2020a). Another common trope was topics related to conception and birth: Mary's immaculate conception and the Virgin's breastfeeding and their impact on the miraculous power of pilgrimage, again with no equivalent in the progressive type (Benedict 2020b).

However, corporeal metaphors were also often used to describe pilgrimage sites, ranging from the stress on the "community of hearts" to which the sacred site gives rise (KTOTV 2020d) to the broadly shared depiction of Lourdes as "the lung of prayer" (Lourdes Rector: Closed Shrine Remains a "Lung of Prayer for the World" 2020). Finally, questions of the gendered body also featured prominently here, linking this type of pilgrimage and the Marian cultus with the conservative political agenda of today. Especially the EWTN focused on this type of connection: "Americans are confused in terms of their identity, their role in the world, what is the role of a father, what is the role of a mother . . . I don't know of any time in the history when we would have needed rosary more than right now" (Power in My Hands the Movie n.d.).

The absence of corporeality in the progressive type of pilgrimage is not accidental as it is tied to an entirely different understanding of what pilgrimage means. Although the materiality of pilgrimage in this type is as important as in the conservative type, here it is expressed in terms of physical movement (of bodies), not the bodies themselves. Relics, such as a saint's remains, healing water, and a blessed souvenir, which are central in the conservative narrative about pilgrimage and its corporeal as well as spiritual benefits, never come up as a topic in the progressive type. Illness and health are almost never discussed in it, and if ever the healing effects of pilgrimage are mentioned, it is only in terms of fitness and the benefits of physical movement for the human body. This is also the reason why the visual focus in the analyzed videos about progressive pilgrimage is not on kneeling or sitting persons, or persons moving within the holy precinct, but on people walking in open ground, in a forest, or in a meadow (KTOTV 2020b).

## 7. Geopolitics of Pilgrimage: The Journey, or the Destination?

As important as the stress on corporeality in conservative pilgrimage and its absence in progressive pilgrimage are, they do not constitute the most important difference between the two. The most fundamental distinguishing element is the way the spatiality of pilgrimage is constructed. For the conservative, Marian type, the pilgrimage is about the destination, which is static, fixed in space: the sacred place, the birthplace of a saint, the shrine, the grotto. Hence, the videos about pilgrimages to Fátima, Lourdes, or Međugorje almost never address the way the pilgrims came to the sites, as if this were entirely irrelevant. The movement from the ordinary to the extraordinary (*separation* in the established Turnerian vocabulary) is not discussed. What pilgrimage means is not *moving* elsewhere, but *being* at the holy place. Hence, when describing a pilgrimage to Lourdes during the COVID-19 times, the typical message is not about *leaving* the secular social structure, but about *being* in Lourdes, as it is "a place of grace, a place of hope and thus also a place of resilience" (KTOTV 2020c). A pilgrimage, in the end, equals being at the holy site, not experiencing movement, change, or liminality.

Another typical feature that confirms this observation is the frequently used analogy between a pilgrimage and a life's journey. This comparison is nigh omnipresent in both types of pilgrimage. However, in the conservative one, the analogy is focused almost exclusively on the destination, the final goal of one's life. The prayers which are used in connection with pilgrimage typically stress the dichotomy between the life on earth as a journey through the valley of tears and the heavenly rewards afterward. A typical example

is the much shared and "liked" prayer by Thomas Kempis from *Catholic and Proud*: "Keep yourself as a pilgrim, and a stranger upon earth, to whom the affairs of this world do not in the least belong. Keep your heart free, and raised upward to God, because you have not here a lasting home." (Catholic and Proud 2020c; but there are many other examples such as Catholic and Proud 2020a).

In the conservative framing, the stories that present a believer's life as a pilgrimage stress the end point, very often in the form of a conversion to the Catholic Church. "The pilgrimage of faith" of one such (originally Protestant and later Anglican) Christian shared on the *Catholic and Proud* Facebook site is a case in point. His life's journey as an Anglican came to a crisis when ordination of women became an important topic in the Anglican Church (Benedict 2016). The answer and solution, which is discussed in great detail, was the conversion to the Catholic faith. The journey serves as a relatively short introduction to the lengthy description and justification of the destination, the Catholic Church. Importantly, the decision is argumentatively linked to two highly politicized markers of conservative/progressive positions in Christianity today: ordination of women and the attitudes toward LGBT+ people.

A similar story appeared on EWTN. In the video, a Catholic author discussed his book *Unlikely Pilgrim* (EWTN 2020c). The structure of the narrative is almost identical to that in the previous example. Originally a Quaker, the author first converted to the Episcopalian Church, but then he found God and himself in the Catholic Church, with the conversion again given ample attention. The book itself explores interesting pilgrimage sites in Europe, once more focusing on places, not the journey. The connection is made to the struggle of the Catholic conservatives against the sinful world, comparing the situation of today with glorious times in the past. The importance of Christianity is "lost on our current generation", and schools forgot to teach people that "what Jesus Christ started was the foundation of the world" (EWTN 2020c).

In the progressive type, on the other hand, pilgrimage is defined as movement, not as a destination. This applies to both the actual pilgrimage and life seen as a pilgrimage. God is not waiting at the end of the journey, but he accompanies the pilgrims on the way: "Living one's life as a pilgrimage, encountering God along the way" (7 Things 2020) is the motto of this type of pilgrimage that was shared by one of the analyzed Facebook sites. If there is a formal destination (such as in the case of the many pilgrimages along the *camino* to Santiago di Compostela), it is not even shown in the video or shown just briefly at the very end (KTOTV 2020b). The focus on movement is then reflected in the attention dedicated to the nature and the physicality of the journey through mountains or forests (KTOTV 2020e). The pilgrims themselves, in their testimonies, confirm that for them, Christianity and pilgrimage are best expressed in terms of metaphors of movement and change, as the aim is to show "the living church, a dynamic church, which always carries the message about Christ" (KTOTV 2020e).

Even the liturgical elements often take place in the open, as people usually pray sitting in meadows, standing in a copse of trees, or while walking along the route (KTOTV 2020b). Not surprisingly, the shift away from the officially designated Catholic places of worship towards moving through the landscape also renders pilgrimage more attractive for those less strongly affiliated with the Catholic Church. This is one of the reasons why the progressive type of pilgrimage is often tied to a more ecumenical orientation and why the Catholic Church or the Catholic faith is seldom explicitly mentioned in this case (KTOTV 2020b).

The overlap between a pilgrimage as a religious ritual and spiritually motivated tourism is present in both of the types of pilgrimage, as the many articles and videos about pilgrimages to Rome or to the Holy Land attest (Catholic and Proud 2020d; Ryan 2017; 7 Things 2020). However, the overlap is much more pronounced with the progressive type. Sometimes, it is even difficult to tell the religious and the touristic aspects apart. For instance, a pilgrimage documentary shared by KTO reiterates the story of St. Augustine and then follows a group of pilgrims on *Via Augustina*. While the Christian focus of

the pilgrimage is clearly visible in the report (KTOTV 2020d), it targets a much wider audience of those interested in "cultural roots" of Europe and the European spiritual heritage, broadly speaking. Those who comment on the pilgrimage in the documentary are not there as religious authorities but as students of Christianity or historians of antiquity. Even the Catholic priest who speaks at length in the documentary is introduced as a historian of Christianity (KTOTV 2020d). Not surprisingly, the pilgrimage is then focused on walking through the North African countryside, and the destination of the pilgrimage is not prioritized over the walk itself.

## 8. Pilgrimage as the Experience of *Communitas*?

One last differentiating element between the two types of pilgrimage is the community-building dimension. The Turnerian interpretation of *communitas* (Turner 1969) rests on the shared feeling of community among fellow humans (for instance, during a rite of passage, but also on a pilgrimage). At the same time, *communitas* works in a dialectical relationship with the normal social structure, nurturing and sustaining it, but also pointing beyond it to the limits of this structure and the possibility of overcoming it. The focus on *communitas* was later criticized by Eade and Sallnow (Eade and Sallnow 2000), who pointed out that pilgrims often bring their own presuppositions and discourses, which are then reflected back at them during the pilgrimage. Hence, Turner's community of pilgrims was contrasted with the individual quest of each pilgrim, which often differs from the ideas, imaginations, and goals of others participating in the same pilgrimage or visiting the same sacred site.

My expectation was therefore that if the community-related aspect were to be found in one of the two types of pilgrimage, it would be the conservative, Marian pilgrimage type. The conservative pilgrimage is typically hierarchically structured, with a well-visible disciplinary power of the Catholic clergy. A (conservative) sacred place is also easier to regulate than a (progressive) route. Hence, I expected a conservatively oriented community-building to be found in the reports on the Marian pilgrimages, with none or very little of it in the progressive type, with its multiplicity of spiritualities and greater diversity of pilgrim backgrounds.

However, the opposite proved to be true. The social media reports about the conservative pilgrimage type are, almost without exception, focused on individual goals of the believers. These goals may differ, as in some cases the goal is the physical healing of the pilgrim or a prayer for somebody else's health, and in others, the central narrative revolves around spiritual benefits, such as deepening of faith. Community is virtually never addressed in these reports, neither in the form of a sense of togetherness among fellow pilgrims nor in terms of strengthening the community from which the pilgrim comes. Even when the prayers do not focus on an individual, they do not target concrete communities, but abstractions such as humanity, the war-afflicted, the suffering, and, recently, those ill with COVID-19 (Virtual Pilgrimage 2020). The storytellers are individuals too, typically Catholic priests or sometimes elderly women (KTOTV 2020c).

For the progressive pilgrimage, on the other hand, the community is absolutely central. In virtually all the reports and videos of this type, we follow groups of people, not individuals (KTOTV 2020b, 2020d, 2020e). They share the same route, jointly prepare food, and eat together. Unlike in the conservative type, the reader is amply provided with intimate details about the group, such as their shared accommodation or camps in nature (KTOTV 2020b). Often, the common purpose of those on the pilgrimage is stressed, such as renovating a church or evangelizing through singing. Even the titles of the documents, such as "Travelling Sowers" (KTOTV 2020b), stress this unified purpose. The typical speakers are young, and often a couple or the entire group are recorded giving their responses together.

## 9. Conclusions

I started my analysis by challenging the still widely respected Turnerian account of pilgrimage. On the one hand, even in an era of virtualization, pilgrimage seems to have

succeeded in retaining its "kinetic", spatial nature. However, the spatiality (or rather spatialities) of pilgrimage is undergoing a fundamental transformation. This transformation is linked to the pluralization of the geopolitics of pilgrimage, the political *and* spatial differentiation of what pilgrimage means and what its aim is. The two types of pilgrimage I identified in the article also challenge the understanding of pilgrimage as a series of three subsequent phases, i.e., separation, liminality, and aggregation.

The conservative type of pilgrimage is based almost entirely on the third aspect, the aggregation. The existing religious order is not to be challenged, but rather reaffirmed. Separation and liminality are suppressed, both in terms of the discourses surrounding the pilgrimage and in terms of the geopolitics of pilgrimage.[7] The Catholic woman who moved to Lourdes and lives there to experience Lourdes permanently (EWTN 2020b) is the ultimate example of this type of devotion. There is no need for separation, no place for a subversive challenge to the established order, and only the aggregation remains.

The progressive type is also closely tied to spatiality, but in a different way. The focus on movement transfers the attention from the *site* to the *journey*. What matters is experiencing the separation and liminality, often expressed as the spiritually enriching hardships of the pilgrimage compared to the comfortable, yet repetitive everyday life. The aggregation is secondary, and so is the destination of the pilgrimage. What matters is being exposed to the transformative potential of the pilgrimage, be it in the form of an encounter with God or a spiritual transformation of the pilgrim's life.

The virtualization of pilgrimage is undoubtedly one of the most visible sociopolitical phenomena accompanying pilgrimages of today. The ascendance of social media and, more recently, the COVID-19 pandemic have, paradoxically, led to the reassertion of the physical, corporeal, and spatial as essential aspects of pilgrimage. Seeing materiality as an essential part of pilgrimage is common to both of the types of pilgrimage that this article analyzes. However, the different ways of defining the spatiality of pilgrimage are markers of two different types of geopolitics of pilgrimage as well. One, focusing on static objects, sites, places, and bodies, extolls the conservative acceptance of the idealized status quo (which is often discursively moved to the past, to a golden age of faith and devotion). The other discards these objects almost entirely and is carried by a strong skepticism towards the static and the unchanging, but also towards the institutional aspect of faith.

This geopolitical division impacts all the dimensions of pilgrimage, ranging from the sites/routes chosen, via the type of devotion exhibited, to the way these two types are presented on social media. It would be therefore a mistake to see the revival of pilgrimage as a revival of a certain narrowly defined spirituality. Instead, we witness two inter-related and yet independent processes, both of which are, confusingly, labeled as pilgrimage. Both of them certainly attest to the vitality of spirituality, or rather, spiritualities in Europe, but they also show that these spiritualities have far-reaching political consequences which often stand at odds with each other.

**Funding:** This work was supported by the Grantová Agentura České Republiky [GA19-10969S].

**Conflicts of Interest:** The author declares no conflict of interest.

## Notes

1　The revival of pilgrimage in Europe has its roots in the combination of the advancing European secularization and the religious reactions to it, the search for nontraditional forms of spirituality, the growth of religious tourism, and a number of others. However, it is interesting to note that a similar trend can be observed across the globe, ranging from the growing popularity of the pilgrimages to Jerusalem to the boom of Hindu mass pilgrimages (including Kumbh Mela in Haridwar this year). Although some of the causes may be identical, this article deals specifically with the revival of pilgrimage in Europe.

2　Within Catholic Mariology, the Age of Mary is a term sometimes applied to the period starting around 1830 when Marian apparitions became much more frequent and the related Marian cultus much more typical for the popular devotion.

3　The study of the Marian devotion and Marian pilgrimage is a narrow, yet thriving subfield of the religious studies. Among the popular studies on the topic, I would like to mention Zimdars-Swarz´s *Encountering Mary. From La Salette to Medjugorje.* (Zimdars-Swarz 2014) and the two great collections of studies edited by Jansen and Hermkens (2009) and by Di Stefano and Ramón Solans (2016).

4   Catherine Labouré was a French nun whose Marian visions led to the creation of the miraculous medal. Even today, wearing the medal belongs to the most distinctive features of traditional Marian devotion.

5   San Giovanni Rotondo is connected with the life of the Catholic saint Padre Pio. Padre Pio was known for his wounds, stigmata, and for his devotion to Virgin Mary, particularly the prayer of rosary.

6   In this article, references are made directly to the shared videos or texts, even if these texts are on a different website, and the Facebook posts only offer links to them.

7   A clarification is needed here. It could be argued that the conservative type of pilgrimage is, in fact, a liminal ritual if what is at stake is the liminality vis-à-vis the society. However, I would argue that in the Turnerian account, there is a deep connection between liminality and the subsequent aggregation. The pilgrim becomes liminal towards her or his community (be it the church or the society) and then becomes reaggregated within the same community. That is obviously not happening in the cases I described above.

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
