# Peer review of "Geopolitics of Catholic Pilgrimage: On the Double Materiality of (Religious) Politics in the Virtual Age"

_religions, doi:10.3390/rel12060443_

Round 1
Reviewer 1 Report
The article analyses the phenomenon of contemporary Catholic pilgrimage, seen in political science perspective, and singles out two very different perspectives on the phenomenon, lebeled conservative and progressive, and marked by different attitudes towards space, and the body.
The contribution is very interesting and innovative, and in my opinion deserves publication, pending some minor changes and additions:
- The word 'geopolitics' usually refers to the spatial underpinnings of interstate relations. In this article it is used in a different and peculiar way, and it is also part of the title. If the authors feel entitled to use it this way, I warmly recommend that they justify this choice in the methodology section of the paper. Otherwise, they might consider a different choice of words.
- Some names and concepts referred to in the paper (apparitional pilgrimage, age of Mary, Catherine Labouré, San Giovanni Rotondo, etc.) are part of the Christian tradition, and are not necessarily familiar to readers from other cultures. A very short explanation would improve the readability of the manuscript.
- Since the revival of pilgrimage is not exclusively a Christian phenomenon (see for example the Shikoku temple pilgrimage in Japan), the authors might briefly explain how their analysis of pilgrimage in the Christian contexts fits in the larger global picture.
Author Response
Dear Reviewer,
Thank you for the constructive comments. I took all your suggestions onboard and made my revisions accordingly. In some cases, I added a paragraph explaining my position (why I use the term geopolitics, which is central to my research programme). In other cases, I added a series of footnotes (for instnace, when explaining who Catherine Labouré was or Padre Pio, etc.).
Thank you again.
Reviewer 2 Report
This is an excellent article and an exciting contribution to the field. The article shows with great clarity that what is subsumed under the term "pilgrimage" consistutes in reality different, though interrelated, practices with different goals and a different attitude to the body, institutions and community.
I have only one minor suggestion for improvement:
Rather than introducing the three phases of pilgrimage by Turner and Turner in the conclusion, the discussion could be slightly expanded into a section or at least be introduced earlier on. This would do more justice to the abstract, where the Turnerian concept of pilgrimage as a three-phase kinetic ritual appears to be quite central. The conclusion could then concentrate on summing up the main research findings.
Author Response
Dear Reviewer,
Thank you for the comments. I revised the article accordingly, moving the Turnerian discussion to the beginning of the article and adding a new paragraph in the conclusion as well.